# Hyperglycemia-Induced miR-467 Drives Tumor Inflammation and Growth in Breast Cancer

**DOI:** 10.3390/cancers13061346

**Published:** 2021-03-16

**Authors:** Jasmine Gajeton, Irene Krukovets, Santoshi Muppala, Dmitriy Verbovetskiy, Jessica Zhang, Olga Stenina-Adognravi

**Affiliations:** 1Department of Cardiovascular and Metabolic Sciences, Cleveland Clinic, 9500 Euclid Avenue NB50, Cleveland, OH 44195, USA; gajetoj@ccf.org (J.G.); krukovi@ccf.org (I.K.); muppals@ccf.org (S.M.); verbovd@ccf.org (D.V.); zhangj4@ccf.org (J.Z.); 2Department of Molecular Medicine, Case Western Reserve University, 10900 Euclid Avenue, Cleveland, OH 44106, USA

**Keywords:** miRNA, tumor-associated macrophages, breast cancer, hyperglycemia

## Abstract

**Simple Summary:**

The incidence of breast cancer is higher in diabetic patients. Cancers of diabetic patients are more aggressive and grow and spread faster than in patients without diabetes. We discovered a novel pathway that is regulated by high blood glucose and promotes breast cancer growth. We have previously studied the details of this pathway and found that it increases the growth of new blood vessels that feed the growing tumor. Our new results presented here suggest that, in addition to the effect on the growth of the cancer blood vessels, the same pathway regulates the inflammation in the tumor. Furthermore, we found that the regulator of this pathway, a small RNA molecule induced by high blood glucose, can be found in blood of animals with breast cancer tumor and thus, can be used as a marker of a tumor. Currently, there are no affordable methods to monitor the recurrence and metastases in breast cancer patients after the removal of the primary tumor. Monitoring miR-467 levels in blood may prove to be a cheap test that can be frequently performed.

**Abstract:**

The tumor microenvironment contains the parenchyma, blood vessels, and infiltrating immune cells, including tumor-associated macrophages (TAMs). TAMs affect the developing tumor and drive cancer inflammation. We used mouse models of hyperglycemia and cancer and specimens from hyperglycemic breast cancer (BC) patients to demonstrate that miR-467 mediates the effects of high blood glucose on cancer inflammation and growth. Hyperglycemic patients have a higher risk of developing breast cancer. We have identified a novel miRNA-dependent pathway activated by hyperglycemia that promotes BC angiogenesis and inflammation supporting BC growth. miR-467 is upregulated in endothelial cells (EC), macrophages, BC cells, and in BC tumors. A target of miR-467, thrombospondin-1 (TSP-1), inhibits angiogenesis and promotes resolution of inflammation. Systemic injections of a miR-467 antagonist in mouse models of hyperglycemia resulted in decreased BC growth (*p* < 0.001). Tumors from hyperglycemic mice had a two-fold increase in macrophage accumulation compared to normoglycemic controls (*p* < 0.001), and TAM infiltration was prevented by the miR-467 antagonist (*p* < 0.001). BC specimens from hyperglycemic patients had increased miR-467 levels, increased angiogenesis, decreased levels of TSP-1, and increased TAM infiltration in malignant breast tissue in hyperglycemic vs. normoglycemic patients (2.17-fold, *p* = 0.002) and even in normal breast tissue from hyperglycemic patients (2.18-fold increase, *p* = 0.04). In malignant BC tissue, miR-467 levels were upregulated 258-fold in hyperglycemic patients compared to normoglycemic patients (*p* < 0.001) and increased 56-fold in adjacent normal tissue (*p* = 0.008). Our results suggest that miR-467 accelerates tumor growth by inducing angiogenesis and promoting the recruitment of TAMs to drive hyperglycemia-induced cancer inflammation.

## 1. Introduction

In the US, breast cancer (BC) is the most common cancer diagnosis in post-menopausal women and is the second leading cause of cancer deaths [1]. Being overweight or diabetic, combined with low physical activity, are the known risk factors for developing BC in women ages 50 and older [2,3]. There are several meta-analyses studying the association between hyperglycemia, diabetes, and BC, however, the exact mechanisms are not well understood [4,5,6]. Overall, the data suggest that patients with hyperglycemia or pre-existing diabetes have a decreased overall survival and disease-free survival [4].

In 2011, inflammation was added to the “Hallmarks of Cancer” as one of the emerging characteristics of cancer reviewed by Hanahan and Weinberg [7]. Chronic inflammation is not only characteristic of obese and T2D patients, but also present specifically in mammary tissue in the majority of obese individuals [7,8,9,10]. This creates an interesting interaction between macrophages and adipocytes and may be important in understanding BC progression [11,12]. In both adipose and mammary tissues from obese individuals, increased pro-inflammatory gene expression (and the characteristic crown-like structures of macrophages) were shown in diet-induced and genetic mouse models of obesity [9,13,14]. Additionally, primary tumors secrete cytokines, such as CCL2, to recruit blood monocytes into the tumor and sustain chronic inflammation. There, they differentiate into tumor-associated macrophages (TAMs) and shift the milieu to support tumor growth [7,12]. Clinically, the increased infiltration of TAMs has a strong correlation with a poor prognosis [15,16].

Thrombospondin–1 (TSP-1) is a matricellular protein that inhibits angiogenesis and promotes the resolution of inflammation [17,18,19,20,21]. Due to the anti-angiogenic properties of TSP-1, its role in cancer growth has been investigated: TSP-1 has been shown to control tumor growth [22,23].

We have discovered a novel hyperglycemia-induced pathway that upregulates miR-467 and, as a result, downregulates its target TSP-1 in a cell–and tissue-specific manner, leading to increased angiogenesis [24]. Injections of a miR-467 antagonist in mice prevented hyperglycemia-induced BC growth and angiogenesis [25]. We have recently found that miR-467 is upregulated by high glucose in cultured macrophages [26]. Here, we report the results of a study investigating the role of a hyperglycemia-induced miR-467-dependent pathway in regulation of inflammation in mouse BC and hyperglycemia models and in hyperglycemic patients. miR-467 was dramatically upregulated in BC tissues of hyperglycemic patients and was found circulating in blood. Therefore, we explored the therapeutic potential of this miRNA and its potential significance as a circulating biomarker of BC tumor.

## 2. Materials and Methods

### 2.1. Experimental Animals and Protocols

All animal procedures were performed according to protocols approved by the Institutional Animal Care and Use Committee and in agreement with the National Institutes of Health Guide for the Care and Use of Laboratory Animals. Animals were housed in AALAC approved animal facilities of the Cleveland Clinic. Wild type C57BL6, BALB/c, *Lepr ^db/db^*, *Dock7 ^m^* + /*Lepr ^db^* (heterozygote control, lean and normoglycemic) were purchased from The Jackson Laboratories.

Animals were sacrificed at end point by exsanguination under anesthesia with ketamine/xylazine (100 mg/15 mg/kg), and organs were collected. Mice used for cell isolation were euthanized by CO_2_ asphyxiation followed by the cervical dislocation.

### 2.2. Induction of Diabetes in Mice

WT C57BL6 and BALB/c mice were given intraperitoneal streptozotocin (STZ; in 20 mmol/L citrate buffer, pH 4.6) injection following the procedure proposed by Jackson Laboratories (50 mg/kg for 5 consecutive days). Age matched controls received citrate buffer injections. Blood glucose was measured starting 48 h after the final STZ injection using the AlphaTrak Blood Glucose monitoring system. Mice with blood glucose >250 mg/dL were selected for the described experiments.

To induce insulin resistance and pre-diabetes, WT C57BL6 mice were fed a Western diet (TD.88137, 40–45% kcal from fat, 34% sucrose by weight, Envigo) starting at 4 weeks of age and throughout the duration of the experiments (~24 weeks of age).

In the mice on Western diet, *Lepr ^db/db^* mice and the control *Dock7 ^m^* + */Lepr ^db^* mice, blood glucose levels were measured at the end of the experiment.

### 2.3. Injection of Cancer Cells and Tumor Collection

EMT6 mouse cells were purchased from American Type Culture Collection (ATCC) and cultured according to ATCC directions. Cancer cells were injected as described in our prior reports. On injection day, cultured EMT6 cells were washed twice with sterile PBS, viable cells were counted, and cancer cells were injected into mammary fat pad (1.5 × 10^6^ cells in 100 μL sterile PBS). Tumors were harvested when the largest tumors in hyperglycemic mice reached the maximum allowed size (1.7 mm^3^) on day 10–14 post injection. Tumors were weighed, frozen in OCT, or processed immediately to isolate RNA.

Blood was collected at the end of the experiment, ~14 days after EMT6 injection and immediately processed for RNA isolation or blood cell isolation/count.

### 2.4. Tissue Samples

Patient’s breast cancer and adjacent normal tissue specimens were obtained from the Cleveland Clinic Tissue Bank and were deidentified. The work was approved by the Cleveland Clinic Institutional Review Board. All patients were female. Patients with HbAc1 < 6 (5.7 ± 0.07) were considered normoglycemic; patients with HbAc1 > 7 (8.7 ± 0.3; *p* = 1.2 × 10^−6^) or documented diabetes diagnosis were considered hyperglycemic.

Discarded blood samples of random hyperglycemic and normoglycemic patients undergoing blood tests on the same day were obtained from the Cleveland Clinic laboratories. Similar to the BC patients, these were assigned to normoglycemic group if HbAc1 < 6 (5.5 ± 0.07) or hyperglycemic group if HbAc1 > 7 (8.8 ± 0.5; *p* = 7.2 × 10^−7^).

### 2.5. Immunohistochemical Staining

Using VECTASTAIN ABC-HRP Kits (Vector Labs, Burlingame, CA, USA), 10 μm sections were stained with antibodies against CD68 (biotinylated clone FA-11, 1:10, AbD Serotec, Oxford, UK), CD31 (1:100, BD Pharmingen), laminin (1:300, Abcam), α-actin (clone ab5694 1:200, Abcam, Cambridge, UK), or anti-TSP-1 Ab4 (clone 6.1 1:100, Thermo, Waltham, MA, USA). Secondary antibodies were included in the species-specific kit, followed by ImmPACT DAB peroxidase substrate (Vector Labs). Slides were scanned using Leica SCN400 or Aperio AT2 at 20× magnification. Positive staining in the images was quantified using Photoshop CS2 (Adobe, San Jose, CA, USA) and Image Pro Plus (7.0).

### 2.6. RNA Isolation and Real-Time Quantitative RT-PCR

RNA was isolated using Trizol reagent (Invitrogen, Carlsbad, CA, USA) and quantified using Nanodrop 2000 (Thermo). To measure miR-467 expression, 1–2.5 μg of total RNA was polyadenylated using NCode miRNA First-Strand cDNA Synthesis kit (Invitrogen) or miRNA 1st strand cDNA synthesis kit (Agilent). Real-time qPCR amplification was performed using SYBR GreenER™ qPCR SuperMix Universal (Thermo) or miRNA QPCR Master Mix (Agilent, Santa Clara, CA, USA). The miR-467 primer (GTA AGT GCC TAT GTA TATG) was purchased from IDT. To measure expression of inflammatory markers, 1–2 μg of total RNA was used to synthesize cDNA using the SuperScript First-Strand cDNA Synthesis System for RT-PCR (Invitrogen). Real-time qPCR was performed using TaqMan primers for Il6, Ccl2, Tnf, Cd68, Cd38, Egr2 (Thermo) and TaqMan Fast Advanced Master Mix (Thermo).

### 2.7. Statistical Analysis

Data are expressed as the mean value ± SEM. Statistical analysis was performed with GraphPad Prism 8 Software. Student’s t-test and ANOVA were used to determine the significance of parametric data, and Wilcoxon rank sum test was used for nonparametric data. A *p* value of <0.05 was considered statistically significant.

## 3. Results

### 3.1. Hyperglycemia Induces EMT6 Cancer Growth

In three different mouse models of hyperglycemia, BC growth was accelerated by the high glucose levels (Figure 1). In a model of Type 1 Diabetes, BALBc mice that were injected with streptozotocin (STZ) to induce hyperglycemia had 2.11-fold increase in tumor weight compared to normoglycemic (citrate buffer injected) controls. (Figure 1a; 0.585 ± 0.2080 g vs. 0.277 ± 0.1213 g, *p* < 0.001). In *Lepr ^db/db^* mice, a genetically diabetic mouse that models Type 2 Diabetes, tumor weights were increased by 4.22-fold compared to their normoglycemic *Dock7 ^m^* + */Lepr ^db^* heterozygote control group (*Lepr ^Dock/db^*) (Figure 1b; 0.373 ± 0.136 g vs. 0.088 ± 0.037 g, *p* = 0.0079). In a diet-induced hyperglycemia model, WT C57BL6 mice fed a “Western diet” for 16 weeks was also used to more closely model chronic hyperglycemia in humans. There was a 2.95-fold increase in tumor weights Western diet-fed mice compared to chow-fed controls (Figure 1c; 0.374 ± 0.157 g vs. 0.127 ± 0.047 g, *p* < 0.001) and had significantly elevated glucose levels (129.4 ± 21.44 mg/dL vs. 86.35 ± 9.172 mg/dL, *p* < 0.001).

We have recently reported increased inflammation in BC tumors of hyperglycemic mice [27,28]: the expression of pro-inflammatory markers *Il6, Ccl2*, and *Tnf* were significantly increased in STZ-treated hyperglycemic mice and *Lepr ^db/db^* mice as compared to normoglycemic control mice. When markers of macrophages (Cd68), pro-inflammatory (Cd38), or pro-resolving (Egr2) macrophages were analyzed, there was a significant increase in Cd68 expression in hyperglycemic mice. Additionally, hyperglycemic mice had higher expression levels of both Cd38 and Egr2 [27,28].

To understand how miR-467 affects macrophage accumulation in BC, a miR-467 antagonist was used in BALB/c mice injected with STZ, and sections of EMT6 tumors were stained with antibodies against markers of macrophages, anti-Cd68 and MOMA-2. Tumors from hyperglycemic mice had a 24.3% increase in miR-467 levels (Figure 2a, *p* = 0.01) and the macrophage infiltration, detected by anti-Cd68 and MOMA-2 staining, was also increased (Figure 2b,c).

In the group that received injections of the control oligonucleotide, we detected a 2.06-fold increase in Cd68 macrophage staining in BC sections (Figure 2b, *p* = 0.03) and a 2.4-fold increase in staining with the MOMA-2 antibody (Figure 2c, *p* = 0.02) in hyperglycemic mice (STZ-treated). In normoglycemic mice, Cd68-positive staining decreased 3.65-fold in response to injections of the miR-467 antagonist *(*Figure 2b, *p* = 0.001), and MOMA-2 staining decreased 2.89-fold *(*Figure 2c, *p* = 0.02). Inhibition of miR-467 with an antagonist, significantly blunted Cd68 macrophage staining in hyperglycemic (STZ-injected) mice by 5.56-fold (Figure 2b, *p* = 0.02) and also decreased MOMA-2 staining by 4.08-fold (Figure 2c, *p* = 0.01).

### 3.2. Expression of miR-467 in Human Breast Tissue Positively Correlates with Glucose Levels 

In collaboration with the Cleveland Clinic biorepository, we obtained normal and malignant breast tissue from chronically hyperglycemic (HbA1c > 7) or normoglycemic (HbA1c < 6) patients. These specimens were assessed for miR-467 expression to determine whether the miR-467-dependent pathway is present in humans. There was a 3.7-fold increase in miR-467 expression in normal breast tissue, comparing hyperglycemic and normoglycemic groups (Figure 3a, *p* = 0.02). In normoglycemic patients, there was no difference in miR-467 levels in malignant vs. normal breast tissue. In hyperglycemic patients, miR-467 expression was dramatically increased 258-fold in malignant tissue when compared to normoglycemic patients (*p* < 0.001) and increased 56-fold compared to normal tissue (*p* = 0.008). Additionally, there was a positive correlation between increasing glucose levels and miR-467 expression in normal breast tissue (Figure 3b, r^2^ = 0.6109).

### 3.3. Increased Macrophage Accumulation and Angiogenesis and Decreased TSP-1 in Human Diabetic BC Samples

In breast tissue from hyperglycemic patients, macrophage staining increased in both the normal tissue (2.18-fold, *p* = 0.04) and malignant tissue (2.17-fold, *p* = 0.002) [28]. To determine whether the negative regulation of TSP-1 by miR-467 is present within the human specimens, breast tissue specimens from chronically hyperglycemic patients with HbA1c > 7 were assessed for TSP-1 protein levels (by an anti-TSP-1 antibody) and levels of the angiogenesis markers (anti-CD31 and anti-α-actin antibodies) (Figure 3c–e). As expected, staining of TSP-1 tended to decrease along with an increase in CD31 staining, a marker of endothelial cells, in hyperglycemic patients compared to the normoglycemic group (Figure 3c,d). α-actin, a marker of smooth muscle cells and maturation of the vessels, was either decreased in normal breast tissue or not changed in tumor tissue of hyperglycemic patients (Figure 3e), suggesting growth of less mature blood vessels in hyperglycemic tissues. In normoglycemic patients, α-actin staining was lower in tumors as compared to the normal tissue, consistent with the less mature leaky blood vessels expected in cancer tissue.

### 3.4. Increased Expression of Pro-Inflammatory Markers and Markers of Macrophages in BC from Hyperglycemic Patients

Similar to mouse tumors, expression of macrophage and pro-inflammatory markers were analyzed in tumors from hyperglycemic patients (Figure 4). Expression of pro-inflammatory markers IL6 and CCL2 were increased in tumors compared to the normal breast tissue (Figure 4a,b). IL6 levels were further increased 5-fold in malignant specimens from hyperglycemic patients compared to the normoglycemic group (Figure 4a, *p* = 0.03). To further understand the type of macrophage population that was present in these tumors, markers of pro-inflammatory (CD38, Figure 4c) and pro-resolving (EGR2, Figure 4d) macrophages were measured. Both populations were increased in tumors. However, there was a 7.5-fold increase in CD38 expression in specimens of tumors from hyperglycemic patients as compared to the specimens of tumors from normoglycemic patients (Figure 4c, *p* = 0.04), but the levels of pro-resolving (EGR2) macrophages marker were decreased 2.4-fold in hyperglycemic tumors (Figure 4d, *p* = 0.006), suggesting that infiltrating TAMs are more pro-inflammatory in human BC tumors of hyperglycemic patients compared to normoglycemic patients.

### 3.5. Level of Circulating Plasma miR-467 Is Higher in Hyperglycemic Patients

We found that miR-467 is a circulating miRNA: we obtained discarded blood plasma samples from both normoglycemic and hyperglycemic patients and quantified the levels of miR-467. Plasma miR-467 in hyperglycemic patients was increased by 80.5% (Figure 5a, *p* = 0.04). The samples were assigned into two groups based on HbA1c levels as described in methods. When correlation with the blood glucose levels in samples was analyzed (Figure 5b), the two groups were clearly different with higher levels of miRNA-467 correlating with increased glucose levels.

### 3.6. Level of Circulating Plasma miR-467 Is Higher in Hyperglycemic Mice

We measured the levels of miR-467 in blood plasma from hyperglycemic mouse models. We did not detect increased plasma levels of miR-467 in STZ-treated *BALB/c* mice (Figure 5c), but plasma from *Lepr ^db/db^* mice had a 2.17-fold increase in miR-467 as compared to normoglycemic *Lepr ^Dock/db^* heterozygote controls (Figure 5d, *p* = 0.03). In a mouse model of diet-induced obesity and hyperglycemia, mice on a Western diet had a 15-fold increase in plasma miR-467 compared to chow-fed mice (Figure 5e, *p* < 0.001).

### 3.7. Level of Circulating Plasma miR-467 Is Higher in Mice with EMT6 BC Tumors

Dramatically increased levels of miR-467 in tumor of hyperglycemic patients and the presence of circulating miR-467 suggested that the levels of circulating miR-467 may be elevated in the presence of a tumor and that miR-467 may clinically useful as a BC biomarker.

We measured miR-467 levels in blood plasma samples from hyperglycemic mouse models injected with EMT6 BC cells. In the mouse group treated with STZ (to induce hyperglycemia), animals with EMT6 tumors had an 85% increase in plasma miR-467 compared to mice without cancer (Figure 5f, *p* = 0.047). Hyperglycemia increased the levels of miR-467 in mice with cancers further: the increased levels were detected in all hyperglycemia models in mice with tumors (Figure 5g–i): e.g., there was a 4.5-fold increase in plasma miR-467 in STZ-injected hyperglycemic mice with tumors compared to the normoglycemic controls with tumors (Figure 5g, *p* = 0.054). In *Lepr ^db/db^* mice with tumors, there was a 46.0% increase in plasma miR-467 compared to normoglycemic mice with tumors, but it did not reach statistical significance (Figure 5h). In WT mice with tumors on the Western diet, plasma miR-467 had a 6-fold increase compared to chow-fed mice with tumors, although it did not reach statistical significance (Figure 5i). We did not detect increased levels of plasma miR-467 in *Lepr ^db/db^* and mice on Western diet (data not shown). We further explored the changes in miR-467 levels in blood cells and bone marrow (BM) as described below, in order to detect any early signs of miR-467-dependent response in mouse models.

### 3.8. miR-467 Levels in Whole Blood, Blood Cells, and Bone Marrow of Mice with BC Tumors

To further understand the source of circulating miR-467 in response to BC, we analyzed fractions of blood for miR-467 in mice on a chow or Western diet. miR-467 levels were significantly upregulated in cellular fraction of blood in the presence of BC (Figure 6a), but when measured in the whole blood (Figure 6b), in isolated red blood cells (RBC) (Figure 6c), or in in white blood cells (WBC) (Figure 6d), no difference was found, suggesting that a minor cellular fraction of blood cells carries miR-467. Lack of changes in miR-467 levels in major blood fractions in response to tumor prompted us to analyze the effects in bone marrow (BM).

### 3.9. Increased miR-467 Expression in Bone Marrow (BM) from Hyperglycemic Mice

BM analysis demonstrated that, in hyperglycemic mice (STZ-injected), there was a 153% increase in miR-467 levels as compared to normoglycemic mice (Figure 7a, *p* = 0.004). A similar effect was observed in mice on Western diet: miR-467 was increased by 74.3% in BM (Figure 7b, *p* < 0.001). In a genetic model of diabetes, Lepr mice, there was no significant differences between the hyperglycemic (*Lepr ^db/db^*) group compared to the normoglycemic heterozygous control (*Dock/db*) (data not shown). In WT mice with EMT6 tumors, we detected a significant increase in miR-467 levels in response to hyperglycemia (Figure 7b). In BM from STZ-treated hyperglycemic mice with tumors, there was a 2.63-fold increase in miR-467 expression compared to BM from normoglycemic mice with tumors (Figure 7a, *p* < 0.001). A similar effect was observed in Western diet-fed mice: in mice with tumors, miR-467 was increased by 40.0% in hyperglycemic mice as compared to normoglycemic mice on chow (Figure 7b, *p* < 0.001).

### 3.10. Increased miR-467 Expression in Bone Marrow from Mice with EMT6 BC Cells

The levels of miR-467 in BM were increased in presence of EMT6 tumor in WT C57BL/6 mice on chow or Western diet a diet-induced hyperglycemia model (Figure 7b): in mice with tumors, the levels of miR-467 were increased 2.7-fold (*p* < 0.001) in the Western diet group and 3.4-fold (*p* < 0.001) in the chow-fed group as compared to mice without tumors (Figure 7b). Although there was no effect of the genetic background on the hyperglycemia-induced BC growth or on the effect of the antagonist (Figure 1) [25,26], the lack of the effect of the tumor on BM levels of miR-467 in BALBc mice (Figure 7a) suggested the effect of a strain on miR-467 levels in BM.

## 4. Discussion

Hyperglycemia (HG) is associated with higher incidence of breast cancer (BC). However, patients with metabolic disorders are uniquely understudied and are often excluded from clinical trials and studies. They represent a large fraction of all BC patients: in our study, 18% of all patients had diagnosed diabetes or multiple high blood glucose test results–twice higher than the average incidence of diabetes in the US population. In the US, 331,530 patients are diagnosed with BC every year. Thus, about 60,000 of the new BC patients per year are also diabetic. Furthermore, even intermittent post-prandial elevation of glucose levels in non-diabetic cancer patients is associated with poorer prognosis. Lowering the glycemic index or the glycemic load of meals led to better outcomes in cancer patients [29,30,31,32,33,34,35,36,37,38,39,40,41,42,43,44,45,46,47,48,49,50,51,52,53]. High blood glucose modulates tissue remodeling, angiogenesis, and inflammation–the events that are intimately related and occur simultaneously in cancer progression. These programs are often regulated through the same or overlapping signaling pathways. Our results indicate that miRNA-467-dependent pathway regulates cancer growth, angiogenesis, and infiltration of TAMs in response to HG. Consistent with the information that even post-prandial elevation of blood glucose influences cancer growth [29,30,31,32,33,34,35,36,37,38,39,40,41,42,43,44,45,46,47,48,49,50,51,52,53], the effect of miR-467 antagonist on TAM infiltration could be observed even in normoglycemic mice. We previously reported that the antagonist inhibits angiogenesis and BC growth in normoglycemic mice of various genetic backgrounds [25,26].

In the last 25 years, since the discovery of miRNAs, it has become clear that these small non-coding RNAs regulate many key steps in physiology and in the development of various pathologies, including BC [54,55,56] miRNAs have been studied as prospective cancer markers and therapeutic agents [55,56]. Using an antagonist of miR-467 to inhibit cancer growth [25,26] may prove to be a helpful method to treat breast cancer progression, especially in triple-negative breast cancers where therapeutic choices are limited.

Currently, there is no safe, reliable, or inexpensive method to detect BC tumors, especially recurrence and metastasis. Mammograms are used routinely, but they do not detect some types of cancer, cannot detect metastasis, and are not without a risk to the patients’ health, which limits their use to once a year to once in two years. MRI is too expensive (and unsafe to use routinely) and ultrasound (or other imaging methods) are not reliable. The current strategy is “wait and see”: oncologists advise to seek medical help if or when symptoms develop. However, by then, the disease has progressed. A test that can be performed routinely to monitor, or even suggest recurrence or metastasis, would be extremely helpful. Our results suggest that miR-467 may prove to be a useful marker of a BC tumor. Even if the marker and the mechanisms studied in this work are only applicable to the diabetic patients, it will still benefit hundreds of thousands of patients.

In this study, we report that miR-467 accelerates tumor growth by promoting the recruitment of TAMs to drive HG-induced cancer inflammation.

Our initial study of miR-467 as a regulator of expression of a potent anti-angiogenic protein thrombospondin-1 (TSP-1) lead to a discovery of pro-angiogenic function of miR-467 in tumors [25,57] that is mediated by downregulation of TSP-1 [26]. Here, we described its effect on TAM infiltration. Although Thbs1^−/−^ mice were not tested in this work (due to a large body of literature describing the functions of TSP-1 in regulation of cancer growth), it became clear from our previous work that miR-467 regulates inflammation not only through the downregulation of TSP-1 but also through other unknown miR-467 targets [27].

We used three mouse models to study the effects of increased blood glucose levels on mouse BC xenograft growth, miR-467 levels, and inflammation: (1) STZ-induced hyperglycemia that imitates human type 1 insulin-dependent diabetes, (2) genetic model of insulin resistance and type 2 diabetes (*Lepr ^db/db^* mouse), and (3) a diet-induced insulin resistant and hyperglycemic model, which may be closest to mimicking the human diet-induced metabolic changes observed in the “Western” population. In all three models, hyperglycemia was associated with the accelerated BC tumor growth. As we previously reported, administration of a miR-467 antagonist prevented the effect of hyperglycemia on tumor size [26]. EMT6 is an epithelial mouse mammary carcinoma cell line [58]. It was established from a transplantable murine mammary carcinoma that arose after implantation of a hyperplastic mammary alveolar nodule and is a model of triple-negative BC. Although a novel therapeutic target to treat BC-associated inflammation (miR-467) is especially valuable for the form of BC that does not have specific and effective therapeutic targets, we previously demonstrated that the effects of miR-467 are not limited to triple-negative BC: in addition to three cell lines of triple-negative BC, we used Ac711 xenograft model [26]. Ac711 is an epithelial origin cell line that was derived from the mammary tumor of a transgenic mouse carrying the Zeta-Glovin-v-Ha-ras transgene and overexpresses Ha-Ras [59].

In addition to mouse models, diabetic patients’ specimens of BC tumors and normal adjacent tissues were examined.

The expression of miR-467 in tumors was upregulated in HG: we have detected a significant increase in EMT6 tumors of three mouse models of hyperglycemia and in hyperglycemic patients’ normal breast and cancer tissue specimens. Interestingly, the expression of miR-467 was also upregulated in the bone marrow of hyperglycemic mice, suggesting that miR-467 may be involved in regulation of production of the inflammatory blood cells or regulation of their functions.

HG was associated with the increased levels of several pro-inflammatory markers, including several cytokines (*Il6, Ccl2*, and *Tnf*) and Cd68 (a marker of macrophages). Immunohistochemistry with anti-macrophage antibodies (anti-Cd68 and MOMA-2) revealed increased numbers of macrophages in tumors of HG mice, and infiltration of tumor-associated macrophages (TAMs) was prevented by administering a miR-467 antagonist. We recently reported that both the markers of pro-inflammatory macrophages (Cd38) and tissue-repair macrophages (Egr2) were upregulated in tumors of hyperglycemic mice suggesting that the polarization of the recruited macrophages was not regulated by miR-467 in the xenograft model [28] However, in hyperglycemic patients’ tissues, there was a significant increase in CD38, a marker of pro-inflammatory TAM, and a significant decrease in EGR2, a marker of anti-inflammatory pro-resolving macrophages. TAM infiltration was higher in both the normal and the cancerous tissues from hyperglycemic patients as compared to specimens from the normoglycemic patients. Furthermore, TAMs were significantly upregulated in tumors vs. adjacent non-cancerous “normal” tissues, supporting more inflammation in tumors, and even further increased infiltration of immune cells in hyperglycemic patients. TAMs contribute to the progression of many cancers [32,39,48,49]. Patients with cancers that have higher TAM numbers have lower survival rates [37]. Clinically, expression of the macrophage marker CD68 is routinely used as a part of the OncotypeDX test that predicts BC aggressiveness and metastasis [60,61]. In a tumor microenvironment, TAMs (together with other leukocytes) secrete a variety of pro-inflammatory factors and promote chronic inflammation and angiogenesis, thus, supporting tumor growth. The inflammation is increased in hyperglycemic patients, and diabetes is associated with a poorer prognosis and a lower disease-free survival of patients with BC [4,5,6,42].

The defining feature of cancer cell is enhanced glycolysis resulting in lactate generation that is called the Warburg effect and is observed even in the excess of oxygen (reviewed in [62]). This metabolic reprogramming is thought to be the most important and even suggested to be responsible for most (if not all) cancer hallmarks [63]. We did observe the effect of miR-467 antagonist on blood glucose clearance in WT mice on chow diet in our previously published study [27], and this observation suggested that miR-467 may regulate influx of glucose into the cells. However, this effect was lost in mice on Western diet where we still see a robust effect of the antagonist on cancer growth, angiogenesis, and TAM infiltration, suggesting that this is not a pathway used by miR-467 to accelerate cancer progression. In turn, the increase in miR-467 does not result from Warburg effect, because it is independent from intracellular glucose metabolism: in various cell types miR-467 was induced by a change in osmolarity [25,27,57]. Although the targets of miR-467 pathway still need to be examined, and the precise molecular mechanisms of hyperglycemia and miR-467 effects need to be uncovered, our results stress the importance of cancer cell interactions with the microenvironment. Thus, exploring the functions of cells (other than cancer cells) is important and may result in new therapeutic targets and markers.

miR-467 levels were higher in hyperglycemic patients’ specimens in both the non-cancerous and cancerous tissues. However, in malignant tissues, the effect of hyperglycemia on miR-467 levels was disproportionally higher, increased by three orders of magnitude as compared to either malignant normoglycemic or non-cancerous tissues from either normoglycemic or hyperglycemic patients. This observation led us to investigate whether miR-467 circulates in blood and whether the presence of a BC tumor increases the levels of circulating blood miR-467. miR-467 was easily detected in human plasma and its levels were upregulated in the plasma of patients with HG. In three mouse models of hyperglycemia (STZ injections, *Lepr ^db/db^*, and Western diet), hyperglycemic mice had higher levels of miR-467 in plasma. When the plasma levels of miR-467 were compared in mice with and without EMT6 tumors, the mice with cancer had higher levels of circulating miR-467, and the levels of circulating miR-467 was consistently higher in hyperglycemic mice with tumors compared to normoglycemic mice with tumors. These data suggest that miR-467 may prove to be a marker of BC in diabetic or chronically hyperglycemic patients. Such marker could be used as an efficient and inexpensive test to monitor metastasis and recurrence after treatment of the primary tumor, similar to the marker that exists for prostate cancer patients. miR-467 circulates in blood and may become a valuable BC tumor biomarker that could be used to detect primary tumor or, more clinically relevant, cancer recurrence and metastatic disease. miR-467 can be easily measured in the blood to indicate primary or to monitor secondary BC tumors.

miR-467 upregulation in response to hyperglycemia is a cell- and tissue-specific phenomenon [25,27,56], but it is not restricted to cancer or blood macrophages only: e.g., we detected increase in miR-467 in the liver of mice on Western diet [27], although the cellular source of miR-467 has not been investigated (hepatocytes versus macrophages versus EC). To understand the source of circulating miR-467 in plasma, we evaluated miR-467 levels in blood cells and bone marrow of mice with tumors. It may be worth mentioning the differences in miR-467 levels in plasma versus BM between the models: STZ-treated mice with tumors had higher levels of miR-467 in plasma, while in mice on Western diet the levels of miR-467 were upregulated in BM and cellular fraction of blood but not in plasma. Although correlation between type 1 diabetes and BC is controversial and poorly addressed, and most clinical studies focus on predominant type 2 diabetes, the reports suggest that type 1 diabetes does not have the same strong connection with BC as type 2 diabetes has [64,65,66]. The difference in expression of miR-467 in two mouse models of hyperglycemia may reflect these clinical observations and could provide clues for the mechanisms of BC growth. miR-467 levels were increased by hyperglycemia in an STZ model. Lack of further increase in miR-467 levels in response to presence of BC tumor may be due to already highest levels of miR-467 (the levels of blood glucose are the highest in STZ model compared to two other models and are similar to uncontrolled diabetes levels in humans), or it may reflect a different fate of the cellular fraction carrying circulating miR-467 (e.g., two models may differ in recruitment of these cells into tumors or the life length of these cells, etc.). Changes in miR-467 levels in BM in response to the EMT6 tumor occurred within days, suggesting that circulating miR-467 may reflect the increased miR-467 levels in specific sub-fractions of BM and the degree of infiltration of tumors with immune cells from BM. When cellular fraction of blood from mice with tumors in the diet-induced model of hyperglycemia was analyzed, we detected increases in miR-467 levels in blood cell fraction, but the analyses of major cell types and plasma did not reflect this increase, again, suggesting that a small, probably BM-derived fraction of blood cells, responds to the presence of a BC tumor by increasing miR-467 levels.

## 5. Conclusions

Our study has documented the effects of HG on BC growth and inflammation. There is currently very little information on the mechanisms by which glucose attracts (and/or retains TAMs) and promotes a pro-inflammatory environment in tumors. Our data suggest that miR-467, upregulated by high glucose, is a regulator of TAMs and cancer inflammation. miR-467 may prove to be a clinically useful marker of a BC tumor and an attractive therapeutic target.

## Figures and Tables

**Figure 1 cancers-13-01346-f001:**
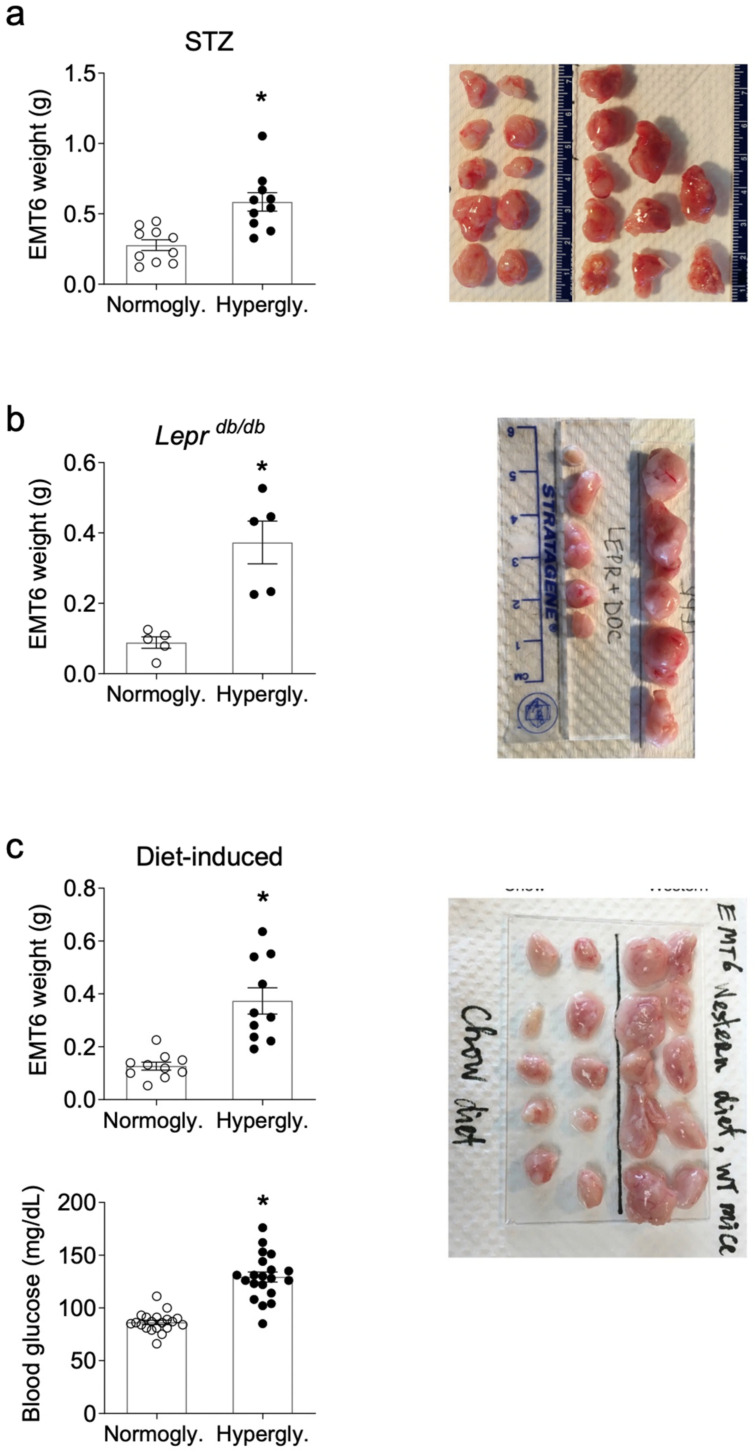
Increased EMT6 BC tumor weights in hyperglycemic mice. EMT6 cancer cells were injected subcutaneously into normoglycemic or hyperglycemic: (**a**) WT mice treated with STZ (*n* = 10/group), (**b**) *Lepr ^db/db^* mice (*n* = 5/group), (**c**) WT mice on Western diet (*n* = 10/group), as described in Methods. Fasting blood glucose levels were measured at end point. Tumors were excised a week later and weighed. Bars represent the mean ± SEM. * *p* < 0.05.

**Figure 2 cancers-13-01346-f002:**
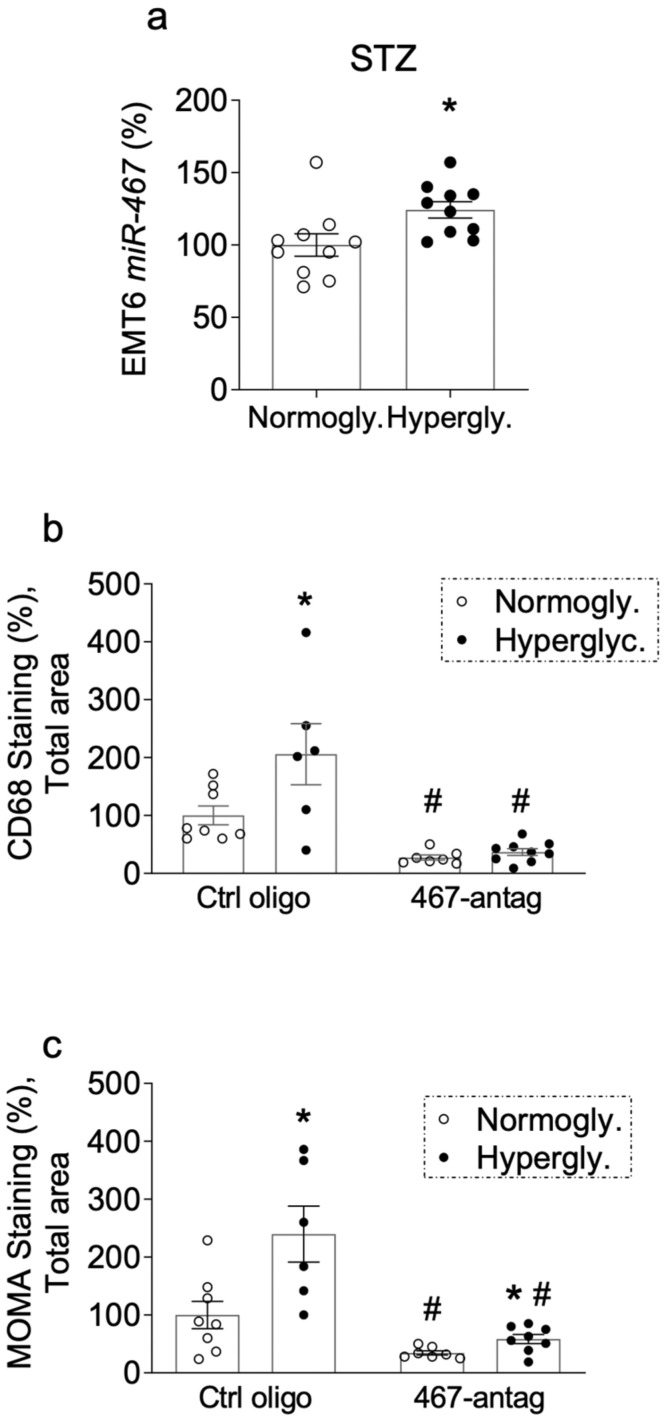
Inhibition of miR-467 with antagonist prevents hyperglycemia-induced macrophage accumulation (**a**) RNA from whole tumors was extracted and miR-467 was assessed (*n* = 10/group). (**b**,**c**) Sections of tumors harvested from hyperglycemic mice (STZ-treated) and normoglycemic mice that received systemic injections of miR-467 antagonist or (**a**) control oligonucleotide were stained with (**b**) anti-Cd68 antibody. (**c**) anti-MOMA-2 antibody. In (**a**–**c**), an increase over the average levels in normoglycemic mice injected with control oligo is shown, with control average as 100% (*n* = 10/group). Scale bars = 10 µM. Bars represent the mean ± SEM. * *p* < 0.05 comparing normoglycemic groups, ^#^
*p* < 0.05 comparing to the group receiving the control oligonucleotide (Ctrl oligo).

**Figure 3 cancers-13-01346-f003:**
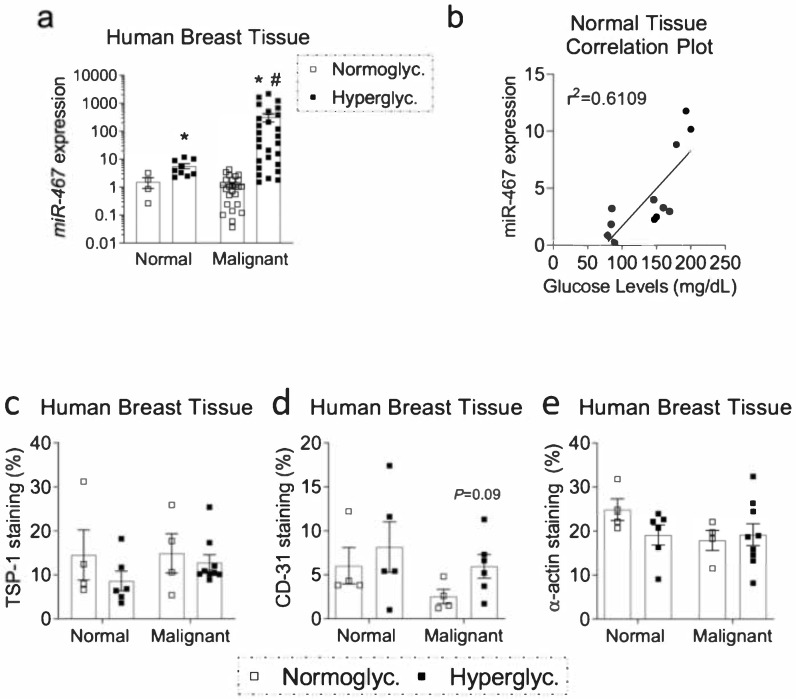
Expression of miR-467 is increased in BC tumors from chronically hyperglycemic patients. (**a**). miR-467 expression was assessed in human breast tissue. Normal = non-malignant adjacent tissue. NB: the *y*-axis is logarithmic. (**b**) Correlation plot: miR-467 expression and glucose levels in adjacent tissue. Sections from human breast tissue were stained for TSP-1 (**c**), and two angiogenesis markers CD31 (**d**) and α-actin (**e**). The % of stained area was quantified per total tissue section area. Bars represent mean  ± SEM. * *p* < 0.05 comparing normoglycemic groups, ^#^
*p* < 0.05 comparing normal tissue.

**Figure 4 cancers-13-01346-f004:**
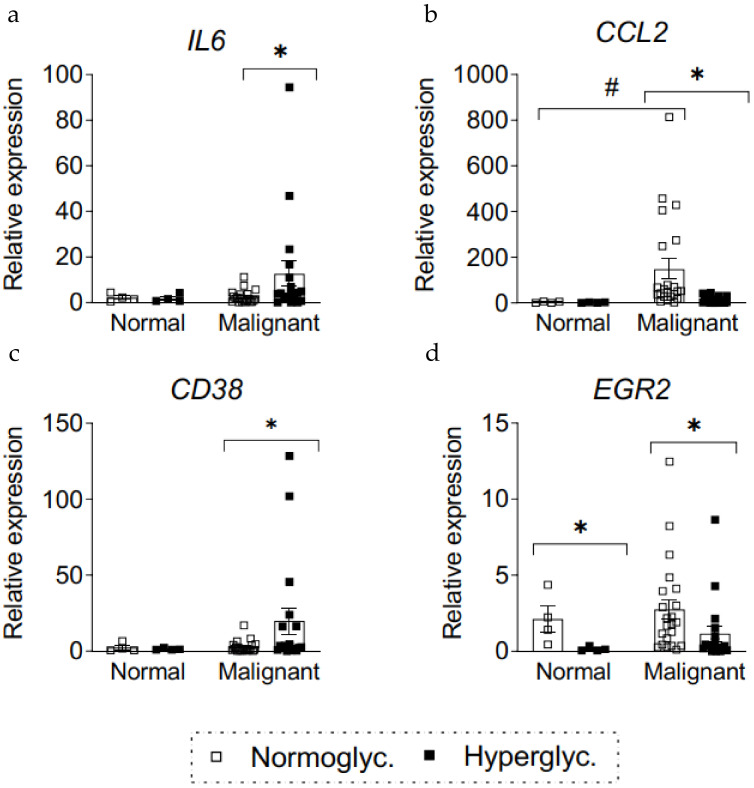
Expression of inflammatory markers in BC tumors from hyperglycemic patients. RNA from whole tumors were extracted and assayed for pro-inflammatory markers (Il6, (**a**); and Ccl2, (**b**) and macrophage markers (Cd38, **c**; and Egr2, **d**) in chronically hyperglycemic or normoglycemic patients. Normal tissue is non-malignant adjacent tissue. Data are normalized to 5S. Bars represent the mean ± SEM. * *p* < 0.05 comparing to normoglycemic controls, ^#^
*p* < 0.05 comparing to normal tissue.

**Figure 5 cancers-13-01346-f005:**
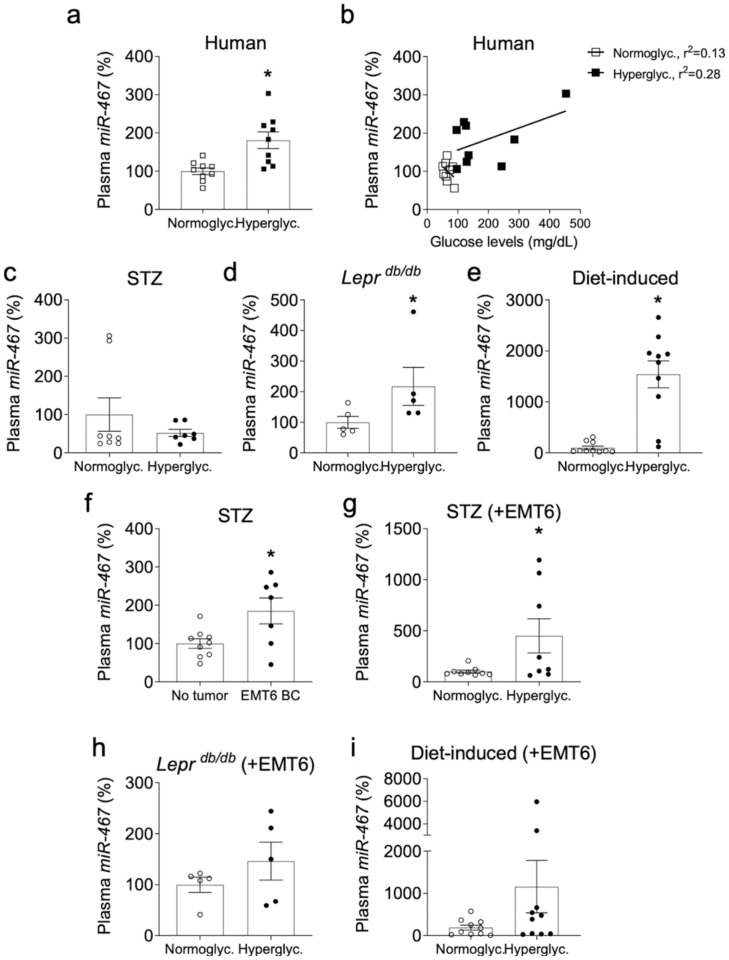
Circulating miR-467 is detected in hyperglycemic patients and mice. (**a**). Plasma from hyperglycemic and normoglycemic patients was acquired from the CCF Biorepository. Plasma miRNA was extracted and miR-467 was quantified by RT-qPCR (*n* = 10/group). (**b**). Correlation between blood glucose levels and miR-467 in plasma. (**c**–**e**). Plasma from hyperglycemic mice was collected from (**c**). STZ-treated, (**d**). *Lepr ^db/db^* mice, and (**e**). mice on Western diet, with corresponding normoglycemic controls. (**f**). Plasma from hyperglycemic STZ-treated mice was collected to compare miR-467 levels in mice with tumors (injected with BC cells EMT6) and control animals without tumors. (**g**–**i**): Plasma was collected from mice with tumors to determine if circulating miR-467 expression is further increased in hyperglycemic mice with tumors: (**g**). STZ-treated mice, (**h**). *Lepr ^db/db^* mice, and (**i**). mice on Western diet. Bars represent the mean ± SEM. * *p* < 0.05 comparing normoglycemic controls.

**Figure 6 cancers-13-01346-f006:**
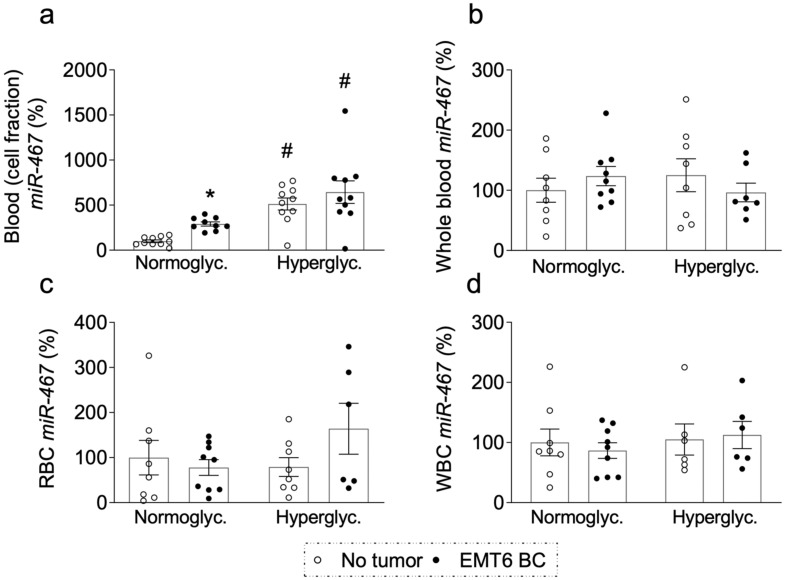
MiR-467 expression in blood of hyperglycemic mice on Western diet. (**a**). miR-467 levels were analyzed in cellular fraction of blood from normoglycemic and hyperglycemic mice with tumors (closed circles) and without tumors (open circles). (**b**). miR-467 levels were analyzed in whole blood from normoglycemic and hyperglycemic mice with tumors (closed circles) and without tumors (open circles). (**c**). miR-467 levels were analyzed in red blood cells (RBC) from normoglycemic and hyperglycemic mice with tumors (closed circles) and without tumors (open circles). (**d**). miR-467 levels were analyzed in white blood cells (WBC) from normoglycemic and hyperglycemic mice with tumors (closed circles) and without tumors (open circles). Bars represent the mean ± SEM, RT-qPCR. * *p* < 0.05 comparing to no tumor. ^#^
*p* < 0.05 comparing normoglycemic controls.

**Figure 7 cancers-13-01346-f007:**
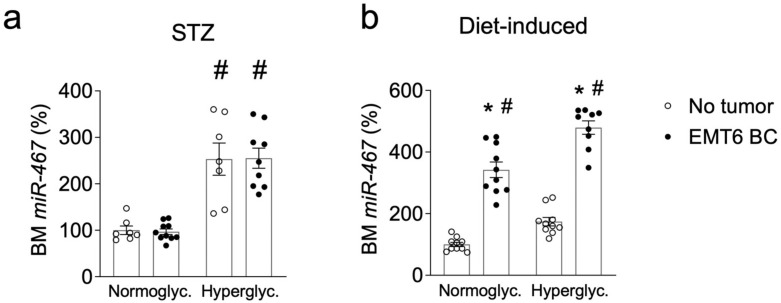
miR-467 expression is increased in bone marrow (BM) in hyperglycemic mice and in C57BL/6 mice with tumors. (**a**). BM was collected from normoglycemic and hyperglycemic (STZ-treated) WT BALBc mice and (**b**). WT mice C57BL/6 mice on Western diet. Bars represent the mean ± SEM. * *p* < 0.05 comparing to the average levels in the control group without tumors. ^#^
*p* < 0.05 comparing normoglycemic controls. Control mice without tumors = open circles; mice with tumors = closed circles.

## Data Availability

All collected data are included in this manuscript.

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
