# Peer review of "Hyperglycemia-Induced miR-467 Drives Tumor Inflammation and Growth in Breast Cancer"

_cancers, 2021, doi:10.3390/cancers13061346_

Round 1

Reviewer 1 Report

This manuscript identifies miR-467 as a marker and potential driver of inflammation and angiogenesis associated in mice and humans with hyperglycemia and with breast cancer. The manuscript is clearly presented, and the data support most of the conclusions drawn with appropriate statistical support. Two issues require additional support.

  1. CD68 is a useful marker for overall macrophage abundance, but their polarization rather than their abundance is the critical issue for understanding their influence on cancer progression. Decreased thrombospodin-1 expression in cancers was previously associated with increased M2 TAMs and decreased numbers and activation of M1 cytotoxic TAMs (PMID: 18757424, PMID: 29986759, PMID: 30537736). Markers of M1/M2 polarization such as NOS2, TNFalpha, and arginase-1 versus -2 should be examined. Does miR-467 alter macrophage polarization?

  1. The finding of increased miR-467 in bone marrow from hyperglycemic mice is interesting, but other organ sources relevant to hyperglycemia and obesity may contribute to the elevation in plasma including adipose, liver or spleen (e g PMID: 33133426, PMID: 31497744). At least one of these other tissues should be examined to establish that the elevation is specific to bone marrow.

Minor issue:

The methods list Thbs1-/- mice, but none of the results presented seem to use those mice. They would be a useful control to evaluate whether TSP1 is a major or minor mediator of the observed responses to miR-467. Were these controls performed?

Reviewer 2 Report

The study of Stenina-Adognravi et al investigates the role of miR-467's association with breast cancer and hyperglycemia. The authors through very elegant experiments show that miR-467 mediates the effects of hyperglycemia and cancer growth. 

The efforts taken by the authors to present this study is appreciated. 

The manuscript is well-written and the data is clearly discussed and the conclusions drawn are concrete. 

I recommend acceptance of this manuscript for publication. 

Reviewer 3 Report

Summary: The authors present their work as a follow-up to other students where they identify miR-467 as being upregulated during hyperglycemia as a stress response in tumors. Here they use chemically induced diabetes in mice and tissues derived from human breast cancer patients to further identify the role of hyperglycemia-induced miR-467, and the pathways that are associated with this change. First, they show in several mice models for diabetes and hyperglycemia that the increase in systemic blood glucose levels causes a similar increase in tumor size from injected mouse breast cancer cells. Next they look at the correlation of miR-467 with markers of macrophages (Cd68), pro-inflammatory (Cd38, IL6, Ccl2), pro-resolving (Egr2) macrophages, TSP-1 that inhibits angiogenesis and promotes the resolution of inflammation, and CD31 as an angiogenesis markers. While the work is exciting, it has certain holes that need to be addressed prior to publication as outlined below.

One of my concerns in the presentation of this article, is that it is often unclear as to what is previously reported work from their lab that they are building up vs new findings. The writers should be careful, particular in the introduction and the result subsections to be clear about what is past work vs what is new work. This is particularly troubling as some of this work isn’t available for public consumption at this point.

This animal was developed into a diabetic model using streptozotocin injections. However, the literature isn’t clear that this would be indicative of Type 2 Diabetes. This is particularly true as Graham et al (Comp Med. 2011) talks about STZ treatment is linked to greater body weight loss which is generally a hallmark of Type 1 Diabetes. Type 2 Diabetes has the greater linkage than Type 1 Diabetes to breast cancer, so I think that this paper needs to focus more on correlating its results with changes to native insulin levels. Was there any measure of Type 2 Diabetes in these mice beyond their blood glucose like weight (BMI), insulin levels, cholesterol, or free fatty acids?

There are different correlations between the different types of breast cancer (by protein marker e.g. hormone positive, HER2 positive, triple negative, etc.) and diabetes. Which type of breast cancer cell is EMT6 most like as it may play a role in the results being reported?

The Patient samples, there is only a discussion of the samples being segregated by HbA1c numbers. Was there any other consideration for factors that they would have diabetes, controlled or not controlled? For example, there is some work that suggests that diabetics being treated with insulin-like or metformin class drugs all have differences in pathology and incidence. This might also suggest there is a difference in the biochemistry behind their breast cancers. Were you told the types of breast cancer that these patients have? You should report the demographics of the cancer type.

The results starts off by talking about results with multiple diabetic models. However, the diet induced hyperglycemic model isn’t mentioned in the methods section. This process should have been described at length there as a second class of diabetes mice models within the methods section. This will also allow for the comparison of the different types of models for different indicators for type 2 diabetes just in the mice themselves.

Fig. 2 – it isn’t clear what is being reported particularly since the values are greater than 100% and the legend says they are “% of stained area was quantified per total section area and multiplied by tumor weight”. What does this mean and how do you then compare them across groups if it has the factor of the tumor weight? Why is the tumor weight here if you are looking at the percentage of the total area?

Fig. 2b,c – is it concerning to the authors that the anti-miR-467 treatment causes a statistically significant decrease in the macrophage markers in the normal glycemic mice?

Fig 4- the authors aren’t consistent in their labeling of the markers for inflammation and macrophage compared to previous text.

Fig 5a- is there any determination as to whether the patient sample demographics from blood plasma mirrored the tissue staining samples.

Fig 5b- these results may be impacted by my earlier comments on the type of diabetes that STZ induces rather than using other mice models that do correlate with human type 2 diabetes. While having cancerous cells may further propagate the effects being seen with miR-467, the drug’s effects cannot be looked over.

Fig. 5e-h – building off the comments from Fig 5b, I would also suggest that the author’s need to consider that this effect is solely being driven by the Warburg effect where cancerous cells have a higher metabolic rate than normal cells.

Fig. 6- what was the time frame between injection and measures? This is particularly important as the preceding paragraph talks about looking for early changes in response to EMT6 tumors. Is there any significance in the RBC cancer on western diet vs the other conditions in part c? It appears to be higher than the other conditions.

Fig. 7- why is the amount of miR-467 in the bone marrow higher in part f than d for Western diets with EMT6? Same question for part h and par d for Chow fed + EMT6? Part e vs part a & c?  This is particularly odd since part g lines up perfectly with the previous parts.

P12- I’m not sure if your fig. 7 data supports using miR-466 as a blood based marker for BC. Matter of fact, your data suggested that the bone marrow would be a better source of discrimination. I’m also not sure if the connection was made with the hyperglycemia and TSP-1 as a marker either.

P13- The authors do not reference the large body of literature about the effect of increased metabolism of glucose in cancerous tissues that might impact the interpretation of their results. This should be included because it gives further credence to the interconnectedness of the tissue and microenvironment for supporting tumor grown.

Round 2

Reviewer 1 Report

The authors have addressed all of my concerns. 

Author Response

We would like to thank the reviewer for the thorough review and positive evaluation of our work.

Reviewer 3 Report

The authors have addressed most of my previous concerns with this manuscript. However, in their revision much of the text for the figure legends have now moved or the correction isn't clear. The authors are asked to revisit the figure legends to make sure they accurately communicate the intended information.

Author Response

We would like to thank the reviewer for the thorough review. We have revised the Figure Legends to correct minor errors (typos, word sequence, etc) and confirm that the Legends are correct and in the right place in the version we are submitting.